# A Soybean Sucrose Non-Fermenting Protein Kinase 1 Gene, *GmSNF1*, Positively Regulates Plant Response to Salt and Salt–Alkali Stress in Transgenic Plants

**DOI:** 10.3390/ijms241512482

**Published:** 2023-08-05

**Authors:** Ping Lu, Si-Yu Dai, Ling-Tao Yong, Bai-Hui Zhou, Nan Wang, Yuan-Yuan Dong, Wei-Can Liu, Fa-Wei Wang, Hao-Yu Yang, Xiao-Wei Li

**Affiliations:** 1College of Life Sciences, Engineering Research Center of the Chinese Ministry of Education for Bioreactor and Pharmaceutical Development, Jilin Agricultural University, Changchun 130118, China; m13630582315@163.com (P.L.); daisiy97@163.com (S.-Y.D.); 18844065895@163.com (L.-T.Y.); 15754366701@163.com (B.-H.Z.); wangnanlunwen@126.com (N.W.); yydong@aliyun.com (Y.-Y.D.); liuweican602@163.com (W.-C.L.); fw-1980@163.com (F.-W.W.); 2Northeast Institute of Geography and Agroecology, Chinese Academy of Sciences, Changchun 130102, China

**Keywords:** *GmSNF1*, salt and salt–alkali stress, soybean, transgenic plants

## Abstract

Soybean is one of the most widely grown oilseed crops worldwide. Several unfavorable factors, including salt and salt–alkali stress caused by soil salinization, affect soybean yield and quality. Therefore, exploring the molecular basis of salt tolerance in plants and developing genetic resources for genetic breeding is important. Sucrose non-fermentable protein kinase 1 (SnRK1) belongs to a class of Ser/Thr protein kinases that are evolutionarily highly conserved direct homologs of yeast SNF1 and animal AMPKs and are involved in various abiotic stresses in plants. The *GmPKS4* gene was experimentally shown to be involved with salinity tolerance. First, using the yeast two-hybrid technique and bimolecular fluorescence complementation (BiFC) technique, the GmSNF1 protein was shown to interact with the GmPKS4 protein. Second, the *GmSNF1* gene responded positively to salt and salt–alkali stress according to qRT-PCR analysis, and the GmSNF1 protein was localized in the nucleus and cytoplasm using subcellular localization assay. The *GmSNF1* gene was then heterologously expressed in yeast, and the *GmSNF1* gene was tentatively identified as having salt and salt–alkali tolerance function. Finally, the salt–alkali tolerance function of the *GmSNF1* gene was demonstrated by transgenic *Arabidopsis thaliana*, soybean hairy root complex plants overexpressing *GmSNF1* and *GmSNF1* gene-silenced soybean using VIGS. These results indicated that *GmSNF1* might be useful in genetic engineering to improve plant salt and salt–alkali tolerance.

## 1. Introduction

Soybeans (*Glycine max*) are one of the most widely grown crops, accounting for approximately 56% of the world’s total oilseed production [1]. However, soybeans are moderately salt-sensitive crops, and salinity stress is one of the challenges limiting soybean production worldwide [2]. Therefore, it is crucial to develop salt-tolerant soybean varieties to meet the global demand for soybean products. Protein kinases are important plant regulators that sense environmental signals through membrane receptor proteins and activate different protein phosphorylation pathways to regulate the expression of downstream resistance genes to protect plants or reduce plant damage from unfavorable environments [3]. In recent years, it has been shown that the sucrose non-fermentable 1-kinase (SnRK1) family regulates plant growth, development, and resistance to saline stress. SnRK1α in tomato significantly upregulated the expression of the salinity stress response genes *SlPP2C37*, *SlPYL4*, *SlPYL8*, *SlNAC022*, *SlNAC042*, and *SlSnRK2*, enhancing salinity tolerance in tomato [4]. SNF1-related protein kinases (SnRKs) comprise three subfamilies: SnRK1, SnRK2, and SnRK3 [5]. SnRK3 kinase, also known as calcium-dependent protein kinase (CIPK), and GmPKS4 of the SnRK3 subfamily showed that GmPKS4 increased tolerance to salt and salt–alkali stresses in transgenic plants [6].

Eukaryotic AMPK/SNF1/SnRK1 protein kinase is a heterotrimeric complex with catalytic α subunit and regulation of β and γ subunits [7,8,9]. There are three genes encoding α subunits, *AtSnRK1α1*, *AtSnRK1α2*, and *AtSnRK1α3*, in *Arabidopsis* [10]. The stomatal index is reduced in the *snrk1α1* mutant of *Arabidopsis*, and its overexpression leads to an elevated stomatal index. *AtSnRK1α1* promotes stomatal development by stabilizing the transcription factor SPCH, which regulates stomatal development [11]. In contrast, overexpression of *AtSnRK1α1* resulted in late flowering and enhanced plant tolerance to nitrogen or carbon starvation, implying their important role in plant growth and stress response [12]. *SnRK1α1* is mainly responsible for the activity of SnRK1 [13]; *SnRK1α2* is rarely reported, while *SnRK1α3* is only expressed at low levels in pollen and seeds [10]. The catalytic α-subunit (SnRK1α1/KIN10 and SnRK1α2/KIN11 in *Arabidopsis*) consists of an N-terminal Ser/Thr kinase structural domain linked to the C-terminal regulatory structural domain, which interacts with the regulatory subunit [14]. Phosphorylation of conserved threonine (T) residues in the catalytic domain “T-loop” is a prerequisite for kinase activity [15]. The catalytic regions of yeast and mammals contain an autoinhibitory sequence (AIS) that inhibits kinase activity [4]. However, in plants, the AIS is not inhibitory and contains a unique ubiquitin-related structural domain (UBA) that mediates its interaction with ubiquitinated proteins [16].

SnRK1 regulates various physiological and biochemical processes in plants and is associated with stress and metabolism [17]. It has been shown that *AtSnRK1α1* can unlearn plant ammonium toxicity and reduce reactive oxygen species (ROS) by controlling the concentration of nitrate at both ends of the cell membrane [18]. Overexpression of *PpSnRK1α* in tomatoes increased ROS scavenging ability, reduced cell membrane damage, and significantly increased antioxidant enzyme gene expression, thereby enhancing plant resistance to salt damage [9]. Germination of transgenic *PeSnRK1a* overexpression seeds under high salt conditions was significantly increased [19], and *GsSnRK1a* overexpression regulates salinity resistance in *Arabidopsis* [20]. Therefore, we hypothesized that the *GmSNF1* gene plays an important role in abiotic stress tolerance also. In this study, GmSNF1 was screened from a yeast cDNA library of soybean roots and leaves under salt and salt-alkali treatment using GmPKS4 as a bait protein with the yeast two-hybrid assay. *GmPKS4* was previously shown to be involved in salt and salt-alkali tolerance in transgenic plants [6]. The interaction of these two proteins was verified using the yeast two-hybrid and BiFC techniques. Its expression pattern and subcellular localization of GmSNF1 were investigated. In a yeast system, *Arabidopsis* plants and GmSNF1-OE soybeans are tolerant to salt and salt-alkali stresses, implying *GmSNF1* is involved in plant responses to these stresses.

## 2. Results

### 2.1. Analysis of the Interaction between GmSNF1 and GmPKS4 Proteins

A yeast two-hybrid assay was performed to confirm the relationship between GmSNF1 and GmPKS4. GmSNF1 has a transcriptional activation domain, and GmPKS4 has a DNA-binding structural domain. The two proteins interact to form a complete transcriptional activator and activate the chromogenic marker. The results showed that co-colonies of GmSNF1 and GmPKS4 showed blue color, and GmSNF1 and GmPKS4 proteins interacted with each other (Figure 1). To further verify that the GmSNF1 protein and GmPKS4 protein interact with each other, a BiFC method based on the transient expression in *Benthamite* tobacco leaves was used. The results showed that intense yellow fluorescence in tobacco leaves co-transformed with pXY104-GmSNF1 and pXY106-GmPKS4, but not in the control, further confirming the interaction between GmSNF1 and GmPKS4 (Figure 2).

### 2.2. Expression Analysis of the GmSNF1 Gene under Abiotic Stresses

A quantitative RT-PCR was used to investigate the *GmSNF1* expression patterns under abiotic stresses in harvested soybean leaf and root samples (Figure 3, Appendix A). The seedlings of soybean were treated with NaCl (salt stress), NaCl + NaHCO_3_ (salt-alkali stress), PEG 8000 (drought stress), or ABA (abscisic acid). The results showed that the expression of the *GmSNF1* gene was significantly upregulated under all four abiotic stresses, more significantly in roots than in leaves (Figure 3). In soybean roots, *GmSNF1* expression was highest at 12 h of salt-alkali stress treatment and then gradually decreased (Figure 3A), whereas *GmSNF1* expression was highest at 6 h of salt stress treatment in soybean leaves (Figure 3B). These results suggested that *GmSNF1* influenced responses to abiotic stresses, especially the salt and salt-alkali stresses.

### 2.3. Subcellular Localization Analysis of the GmSNF1 Protein

The pGDG-GmSNF1-GFP plant expression vector was constructed by inserting the full-length *GmSNF1* cDNA sequence, without a stop codon, into the pGDG-GFP vector. The pGDG-GmSNF1-GFP recombinant plasmid and empty pGDG-GFP vector (control) were infiltrated into young leaves of 4-week-old *Benthamite* tobacco plants. The signal of GmSNF1-GFP accumulated mainly in the nucleus and potentially in the cytoplasm, as revealed by green fluorescence. The results indicated that GmSNF1 was mainly localized in the nucleus and cytoplasm (Figure 4).

### 2.4. Heterologous Expression of the GmSNF1 Gene in a Yeast System Enhances Salt and Salt–Alkali Tolerance of Yeast Strains

The full-length *GmSNF1* cDNA sequence was inserted into the pYES2 vector using the *Bam*HI and *Eco*RI restriction sites to construct the pYES2-GmSNF1 plant expression vector, which was then transformed into the yeast receptor INVSc1. Synthetic complete medium (SC medium) is known as a yeast basal medium. Yeast strains were spotted on salt stress (SC + 500 mM NaCl) and salt–alkali stress (SC + 490 mM NaCl + 10 mM NaHCO_3_) media, and their growth was observed. The yeast strains transfected with the soybean *GmSNF1* gene grew better on salt and salt-alkali-stressed media than the yeast strains transfected with the pYES2 control (Figure 5). These results indicate that the *GmSNF1* gene enhanced the salt and salt-alkali tolerance in yeast.

### 2.5. GmSNF1 Overexpression Enhances Salt and Salt–Alkali Stress Tolerance in Arabidopsis Plants

For the functional analysis of *GmSNF1*, six independent transgenic lines (T1) were obtained by generating *Arabidopsis* plants that overexpressed *GmSNF1* using the floral-dip method. The high-expression lines were screened using qRT-PCR, and the OE-2 and OE-5 lines showed higher *GmSNF1* expression (Appendix A); therefore, these two lines were selected for subsequent soil phenotype experiments. In this study, we found that the leaves of OE-2 and OE-5 were hardy and still green after salt and salt-alkali stresses, whereas the leaves of the wild type were purple and yellow (Figure 6A). The results of physiological indicators showed that the catalase (CAT) activity both OE-2 and OE-5 were higher than those of the wild type, and the hydrogen peroxide content of the wild type was higher than that of transgenic lines, indicating that transgenic lines were less damaged (Figure 6B,C). These results indicate that the soybean *GmSNF1* gene enhances salt and salt–alkali resistance in *Arabidopsis*.

### 2.6. GmSNF1 Overexpression Enhances Salt and Salt-Alkali Stress Tolerance in Soybean Transgenic Hairy Root Complex Plants

To further evaluate the salt and salt-alkali resistance function of the *GmSNF1* gene in soybean, an *Agrobacterium tumefaciens* transfection method was used to generate overexpressing *GmSNF1* soybean complex plants (GmSNF1-OE), which were analyzed by qRT-PCR to quantify *GmSNF1* transcription. As expected, the expression of the *GmSNF1* gene was higher in GmSNF1-OE soybean than in the vector control (VC) (Appendix A). Transgenic soybean plants and vector controls were subjected to salt (150 mM NaCl) and salt-alkali stresses (110 mM NaCl + 40 mM NaHCO_3_). Under these stresses, the leaves of GmSNF1-OE soybeans were tender and green, and the leaves of VC plants were more wilted (Figure 7A). Physiological indicators showed that the GmSNF1-OE soybeans resulted in less hydrogen peroxide and higher CAT enzyme activity compared to the VC plants (Figure 7B,C). The results indicated that overexpression of *GmSNF1* in the soybeans can reduce damage, and the *GmSNF1* gene enhanced the salt and salt-alkali tolerance of soybeans. Soybean salinity tolerance traits are mediated by salinity tolerance genes or by increasing the activity of related antioxidant enzymes. Some salt-alkali tolerance genes *GmPKS4*, *GmSNF4*, and *GmERF7* were selected for analysis with a qRT-PCR assay. In combination with physiological indicators, the CAT activity and hydrogen peroxide content of GmSNF1-OE soybeans and VC plants were significantly different; the CAT enzyme synthesis gene *GmCAT* was selected also. ABA is a well-known stress hormone that plays a key role in the adaptive responses of plants to various abiotic stresses, such as salt and salt-alkali stress. MpSnRK2.10 confers salt stress tolerance in apple via the ABA signaling pathway [22]. A MYB-related transcription factor from peanut, AhMYB30, improves freezing and salt stress tolerance in transgenic *Arabidopsis* through both DREB/CBF and ABA-signaling pathways [23]. Two key genes of ABA biosynthesis, *GmABI1* and *GmABI2*, were evaluated to reflect the response of ABA in transgenic plants. The expression of all the above genes was significantly higher in GmSNF1-OE soybeans than that in the control plants, and the relative expression of the *GmCAT* gene was the highest (Figure 8 and Appendix A). These results suggest that the *GmSNF1* gene upregulates the transcription of stress signal-related genes under salt and salt-alkali stress.

### 2.7. GmSNF1 Gene Silencing Enhances the Sensitivity of Soybean to Salt and Salt-Alkali Stress

To further demonstrate the salt and salt-alkali tolerance function of the *GmSNF1* gene, the TRV-VIGS vector was constructed by selecting a specific 300 bp sequence in the CDS region of the *GmSNF1* gene, and the soybean *GmSNF1* gene-silenced lines were identified using qRT-PCR. The expression of the *GmSNF1* gene in the gene-silenced lines was approximately 0.28-fold higher than that under null load (Appendix A). The *GmSNF1* gene-silenced lines and the vector controls were subjected to stress treatments with 200 mM NaCl (salt stress) and 160 mM NaCl + 40 mM NaHCO_3_ (salt-alkali stress). Phenotypic differences between the *GmSNF1* gene-silenced and vector control plants appeared at around 10 days of stress, with more severe leaf yellowing and wilting in the gene-silenced plants than in the vector control plants (Figure 9A). The results of the physiological indicators showed that CAT activity of *GmSNF1* gene-silenced plants was lower than that of the vector control plants, and hydrogen peroxide content was higher than that of the vector controls under stress conditions (Figure 9B,C). This indicates that the *GmSNF1* gene-silenced plants were more affected than the vector control plants and that *GmSNF1* gene silencing inhibited soybean resistance to salt and salt-alkali stress.

The expression of salt–alkali tolerance genes, CAT enzyme synthesis genes, and ABA pathway gene transcript levels in the *GmSNF1* gene-silenced plants was lower than that in the vector control plants, and the expression of *GmCAT* was the lowest (Figure 10 and Appendix A). This contrasts with the results obtained from GmSNF1-OE soybeans. These results suggest that silencing the *GmSNF1* gene in soybean affects the transcription of genes related to stress signal transduction and CAT enzyme activity, resulting in the sensitivity of soybean to salt and salt–alkali stress.

## 3. Discussion

Under saline stress conditions in plants, the calcium sensor protein SOS3 (also known as calcium-regulated phosphatase B-like 4, CBL4) senses and binds Ca^2+^ and subsequently forms an SOS3-SOS2-SOS1 signaling network with the protein kinase SOS2 (also known as CBL-interacting kinase 24, CIPK24) to enhance salinity stress tolerance in plants [24]. GmPKS4 belongs to the CIPK family, also known as CIPK6, and we had experimentally demonstrated that the *GmPKS4* gene has a salt and salt-alkali tolerance function [6]. In this study, GmSNF1 was screened from a yeast cDNA library of soybean roots and leaves under salt and salt-alkali treatment using GmPKS4 as a bait protein with the yeast two-hybrid assay. With an aim to understand the interaction between GmSNF1 and GmPKS4, yeast two-hybrid and BiFC techniques were performed to verify the result. Similar functional analysis has also been described among other SNF1 homologs. An experiment showed that the C-terminus of apple MdSNF1 interacts with MdJAZ18 to regulate sucrose-induced anthocyanin and proanthocyanidin accumulation in apple [25]. The C-terminus of AtKIN10 (SNF1 homolog in *Arabidopsis*) interacts with FUSCA3 to control lateral organ development and phase transition in *Arabidopsis* [26]. The apple MdSNF1, *Arabidopsis AtKIN10*, and soybean *GmSNF1* are highly structurally conserved; therefore, the sites where they interact may be the same. And it is hypothesized that the GmSNF1 protein interacts with the GmPKS4 protein to regulate salt and salt–-alkali tolerance in plants, suggesting that GmPKS4 is involved in GmSNF1 signaling, and the GmSNF1 protein may play an essential function in salt and salt-alkali stress also.

Sucrose non-fermentable 1 (SNF1)-related protein kinase (SnRK) is essential for regulating plant growth, development, and stress responses [27]. The SnRK1/AMPK/SNF1 kinase regulates cellular carbon metabolism, including glucose utilization in yeast and mammals, ATP production in mammals, and sucrose utilization in plants [28]. SnRK1 has been studied in more detail in plants, mainly in *Arabidopsis* and, to a lesser extent, in soybean. In the present study, we analyzed its expression under abiotic stress conditions and found that the *GmSNF1* gene responded positively to ABA, drought, salt, and salt–alkali stress, suggesting that the *GmSNF1* gene may be played as a linker of the multiple stress signaling pathways. Scientists have analyzed the expression pattern of the *SiSNF1* gene in cereals under abiotic stress in the past few years. The expression of the *SiSNF1* gene in grains gradually increased after salt stress treatment, with the highest expression at 9 h of salt treatment and then gradually decreased [29]. And the result was different from that of the *GmSNF1* gene in soybean, which was induced highest at 12 h under salt–alkali treatment in roots and at 6 h under salt treatment in leaves. This showed that plant *SNF1* genes may be differentially expressed between dicotyledons and monocotyledons. Expression analysis of the soybean *GmSNF1* gene under salt and salt–alkali stress laid the foundation for subsequent plant stress-resistance studies. Successful cloning of the *GmSNF1* gene and subcellular localization analysis of the GmSNF1 protein revealed that the GmSNF1 protein was mainly localized in the nucleus and cytoplasm. Experiments have been conducted to link the function of development-related transcription factors to the nuclear localization of SnRK1α, revealing the specific function of SnRK1α as a regulator of nuclear gene expression [30,31,32,33]. It has also been found that sugarcane ShSnRK1α is localized in the nucleus and cytoplasm of rice leaf pulp protoplasts, thus regulating sucrose accumulation in sugarcane [34]. This is consistent with the results of the subcellular localization of the GmSNF1 protein in this study, suggesting that the GmSNF1 protein might function mainly in the nucleus also. It has been reported that SnRK1α protein is localized to the endoplasmic reticulum (ER) in the validation of FLZ protein interactions with SnRKα protein, indicating that SnRK1α protein is dynamically localized between the nucleus and the ER. Under physiological conditions, SnRK1α is partitioned between the nuclear and non-nuclear fractions associated with the ER. This dual distribution is affected when plants are treated with inhibitors to interfere with plastid energy production, supporting a role for SnRK1α as a sensor of cellular energy status, integrating signals from the chloroplast [35]. Therefore, it is hypothesized that GmSNF1 protein exercises its function mainly in the nucleus.

The advantages of rapid growth and ease of genetic manipulation in yeast combine with the relevance of eukaryotic expression systems [36]. The *Chlorella vulgaris ChACBP* gene is involved in saline tolerance, and the transgenic *ChACBP* yeast strain grew better than the control, indicating that *ChACBP* enhanced the salinity tolerance of the yeast strain [37]. Heterologous expression of SOS-related genes in *Amaranthus marmoratus* confers salt tolerance in yeast [38]. Therefore, this study was conducted to validate the salt and salt–alkali resistance function of *GmSNF1* in a yeast system. Deleting the yeast *SNF1* gene increases the sensitivity of yeast cells to Na^+^, alkaline pH, oxidative stress, heat shock, and genotoxic stress, suggesting that *SNF1* may act as a positive regulator of resistance to these stresses in yeast [39]. Consistent with this observation, the yeast strain trans-GmSNF1 grew better than the control strain under salt and salt–alkali stresses, suggesting that GmSNF1 shares significant structural and functional similarities with its direct homolog, yeast SNF1 and mammalian AMPK. In this study, the yeast strain trans-GmSNF1 was resistant to salt and salt–alkali stresses, indicating that the soybean *GmSNF1* gene is salt–alkali-tolerant.

Excess reactive oxygen species (ROS) can damage biological macromolecules and exert toxic effects on cells. The antioxidant enzyme system plays key roles in plant ROS [40]. CAT is an important protective enzyme system in plants and is the most important enzyme for scavenging reactive oxygen species [41]. Salt and salt–alkali stress can lead to a large accumulation of ROS and inhibition of the antioxidant system, damaging the integrity of cell membranes and plant growth [42]. Numerous studies have shown that the salinity tolerance of plants is enhanced by increasing the activity of antioxidant enzymes in plants. For example, overexpression of *Chrysanthemum* CmCIPK8 regulates salt tolerance in chrysanthemum by increasing the activity of antioxidant enzymes [1]. This is consistent with the results of the present study, in which the overexpression of the *GmSNF1* gene in Arabidopsis and in GmSNF1-OE soybeans were tender green and had higher CAT enzyme activity and lower hydrogen peroxide content than the control plants under salt stress and salt–alkali stress. However, the soybean *GmSNF1* gene-silenced lines were more wilted than the vector control plants and had lower CAT enzyme activity and higher hydrogen peroxide content. This indicated that the overexpression of the *GmSNF1* gene in *Arabidopsis* and in GmSNF1-OE soybeans was less damaged by stress, whereas the *GmSNF1* gene-silenced lines were more damaged than those in wild-type plants. Expression analysis of the *GmCAT* gene was performed in combination with CAT enzyme activity changes in transgenic soybeans, and *GmCAT* gene expression was increased in GmSNF1-OE soybeans and decreased in *GmSNF1* gene-silenced soybeans. Some stress-related genes *GmSNF4*, *GmPKS4*, *GmERF7* and key genes of ABA biosynthesis *GmABI1* and *GmABI2* were also analyzed. The results showed that the expression of the above genes was significantly higher in the GmSNF1-OE soybeans than in the control and lower in the GmSNF1 gene-silenced plants than in the control. The results are similar to other studies. It was shown that overexpression of *GsERD15B* in soybean increased the transcript levels of *GmCAT*, *GmABI1*, and *GmABI2* and enhanced salt tolerance in soybean [43]. Overexpression of *GmERF7* enhances salt tolerance in tobacco plants [44]. It has been shown that *PpSnRK1α* enhances ROS metabolism and salt tolerance in tomato by increasing the expression level of antioxidant enzyme genes and antioxidant enzyme activity [4]. This is consistent with our experimental results, where the expression of the *GmCAT* gene was the most significantly changed among the above genes, indicating that the soybean *GmSNF1* gene enhances the salt and salt–alkali tolerance of plants by upregulation the expression of the *GmCAT* gene and elevating CAT enzyme activity. This study provides new insights into the mechanism of *GmSNF1* regulation in the soybean abiotic stress response and identifies *GmSNF1* as an ideal candidate gene for improving salt and salt–alkali tolerance in soybean.

## 4. Materials and Methods

### 4.1. Yeast Two-Hybrid Validation

The coding regions of *GmPKS4* and *GmSNF1* were amplified; ligated into the pGBKT7 and pGADT7 vectors, respectively; and transfected into the yeast strain Y2HGold (Ang Yu Bio, Shanghai, China). Transfected cells were grown on SD/-Ade/-His/-Leu/-Trp/X-α-Gal medium (ELITE-MEDIA, Shanghai, China) for interaction testing.

### 4.2. BiFC Verification

The coding sequences of *GmPKS4* and *GmSNF1* were cloned into the pXY106 and pXY104 vectors and transferred into the *A. tumefaciens* receptor GV3101 (Ang Yu Bio, Shanghai, China), transiently expressed in *Benthamite tobacco*, and YFP fluorescence was analyzed using confocal laser scanning microscopy with excitation and emission wavelengths of 488 and 507 nm, respectively, to observe the tobacco leaf flesh cells.

### 4.3. Plant Materials

After 14 days of hydroponics, the plants were divided into four groups and treated with 10 mM PEG8000 (BIO FROXX, Beijing, China) to simulate drought stress, 120 mM NaCl (Sinopharm Chemical Preparation Co., Shanghai, China) treatment, simulating salt stress; 5 µM ABA treatment, simulating ABA stress; and 70 mM NaCl and 50 mM NaHCO_3_ (Sinopharm Chemical Preparation Co.) treatment, simulating salt and salt–alkali stress. Samples were collected at 0, 1, 3, 6, 12, 24, and 48 h and kept at −80 °C. RNA was extracted from plant roots and leaves using an RNA kit (Dyna Science Biologicals, made in Beijing) and reverse transcribed into cDNA for real-time fluorescence quantitative PCR (allometric gold reagent) to analyze the stress-induced tissue-specific expression of *GmSNF1* in soybean (Appendix A).

### 4.4. qRT-PCR

*GmSNF1* expression patterns in soybean leaves and roots were analyzed using qRT-PCR, PerfectStart Green Premixes, and a real-time fluorescent quantitative PCR system (Applied Biosystems, Foster City, CA, USA). RNA was extracted and reverse transcribed into cDNA (All-Formula Gold Reagent, Changchun, China) using a Dyna-Co Bio Kit (Dyna-Co, Changchun, China). qRT-PCR was completed to analyze the expression of *GmSNF1* and defense-related genes in the GmSNF1-OE soybeans and *GmSNF1* gene-silenced lines. Melting curve validation analysis was performed using the following cycling parameters: 94 °C for 30 s, 94 °C for 5 s, and 60 °C for 30 s, with 40 cycles. Detailed information on the qRT-PCR primers is listed in Appendix A. The qRT-PCR was performed using three biological replicates. Data were analyzed using the 2^−ΔΔCt^ method [45].

### 4.5. Cloning and Subcellular Localization Analysis of the GmSNF1 Gene

In this study, the sequence of the *GmSNF1* gene was obtained from the plant genome database Phytozome (https://phytozome-next.jgi.doe.gov, accessed on 20 September 2020), *GmSNF1* cDNA was amplified using PCR, the PCR product was purified, inserted into the pMD18-T cloning vector (Mona, Beijing, China), and sequenced to confirm the accuracy. The pMD18-T-GmSNF1 bacterial broth was used as a template to construct the pGDG-GmSNF1 plant expression vector via seamless cloning, and the correctly sequenced broth was transferred to the sensory state of *A. tumefaciens* GV3101. The backs of healthy *Benthamite* tobacco leaves were injected with bacteriophage until the bacteriophage flowed out, labeled well, and placed under a laser confocal microscope for 48–72 h. The fluorescence signal was detected under a laser confocal microscope with excitation and emission wavelengths of 488 and 507 nm, respectively, to observe the tobacco leaf flesh cells.

### 4.6. Heterologous Expression of GmSNF1 in the Yeast System

The *GmSNF1* gene was cloned into the yeast expression vector pYES2, transferred into the yeast receptor INVSc1(Culebra, Beijing, China), and spotted on the control, salt-stressed, and salt–alkali-stressed media to observe strain growth.

### 4.7. Overexpression of the GmSNF1 Gene in A. thaliana and Analysis of Stress Tolerance

The *GmSNF1* gene was cloned into the pCAMBIA3301 plant expression vector and transferred into *A. tumefaciens* receptor state EHA105 (Ang Yu Bio, Shanghai, China). *A. thaliana* was infiltrated using the flush-dip method. The seeds collected from the infiltrated *Arabidopsis* were sown, and when the *Arabidopsis* reached the four-leaf stage, they were sprayed with 1% Basta on the first, third, and fifth days for a total of three sprays. *Arabidopsis* plants that did not wilt or were vigorous were transferred to a new cassette, and genome extraction was performed for identification. The T2 generation of transgenic *Arabidopsis* was spotted onto MS medium (Phyto Tech, Lenexa, KS, USA) containing Basta resistance, and when the seeds germinated, the ratio of the number of germinated to ungerminated transgenic *Arabidopsis* was counted to determine whether it matched 3:1. The T2 generation seeds were sown, and when the rosette leaves of *Arabidopsis* were large enough and vigorous enough, the *Arabidopsis* leaves were cut, and RNA was extracted, reverse transcribed into cDNA, and used as a template for fluorescence quantitative PCR. The highly expressed lines were screened out and cultured, and the T3 generation seeds were collected for phenotypic experiments.

Seeds of the collected high-expression line T3 generation were sown, and after three pairs of compound leaves grew, uniformly growing *Arabidopsis* was transplanted into a square pot. The seedlings were subjected to salt stress (100 mM NaCl, 150 mM NaCl, and 200 mM NaCl) and salt–alkali stress (20 mM NaHCO_3_ + 80 mM NaCl, 30 mM NaHCO_3_ + 120 mM NaCl, and 50 mM NaHCO_3_ + 150 mM NaCl) treatments for about 15 days, and phenotypic changes were observed.

### 4.8. Stress Tolerance Analysis of Soybean Hairy Root Complex Plants Overexpressing the GmSNF1 Gene

The pCAMBIA3301-GmSNF1 vector was transferred into the *Agrobacterium* K599 (Solebel, Beijing, China), and soybean cotyledons emerging from vermiculite (3 cm) were injected. When germinating roots started to form at the infection site, they were covered with vermiculite, and humidity was maintained by adding Hoagland’s solution. After two weeks, when the hairy roots grew strongly, the main roots were removed, and the plants were transferred to pots containing vermiculite. Using the qRT-PCR technique, the overexpressed *GmSNF1* gene hairy roots were identified, and the lines were named GmSNF1-OE soybeans. Approximately three days later, control and GmSNF1-OE soybeans were transferred to Hoagland solution and subjected to salt stress (150 mM NaCl) and salt–alkali stress (110 mM NaCl + 40 mM NaHCO_3_).

### 4.9. VIGS Technology Silences GmSNF1 Gene

The TRV-VIGS vector was constructed by selecting a specific sequence of 300 bp in the CDS region of the *GmSNF1* gene. The *GmSNF1* gene was cloned into the vector pTRV2 and transferred into the *A. tumefaciens* receptor GV3101. And the gene-silenced lines were identified by qRT-PCR and are named soybean *GmSNF1* gene-silenced lines hereafter. Soybean *GmSNF1* gene-silenced lines were identified using qRT-PCR. The silenced lines and vector controls were subjected to stress treatments with 200 mM NaCl (salt stress) and 160 mM NaCl + 40 mM NaHCO_3_ (salt–alkali stress). Phenotypic differences between the *GmSNF1* gene-silenced and vector controls appeared at around 10 days of stress.

### 4.10. Stress-Related Gene Expression Analysis

GmSNF1-OE soybeans and *GmSNF1* gene-silenced plants were watered with Hoagland solution containing NaCl or NaCl + NaHCO_3_ until phenotypic differences were observed. Leaves of the gene-silenced plants and roots of the GmSNF1-OE plants were cut for RNA extraction. In this study, the expression levels of *GmPKS4*, *GmSNF4*, *GmCAT*, *GmABI1*, *GmABI2*, and *GmERF7* were analyzed in soybean hairy root complex plants and *GmSNF1* gene-silenced plants.

### 4.11. Measurement of Physiological Indicators

An activity assay kit (Jiancheng, Nanjing, China) was used to evaluate the treatments of *Arabidopsis* plants, GmSNF1-OE soybeans, and gene-silenced plants for CAT and H_2_O_2_
*Arabidopsis* physiological indicators were determined by mixed sampling of transgenic lines and wild-type lines, respectively.

### 4.12. Statistics Analysis

All experiments were performed in triplicate. Grimer8 was used to prepare the graphs. Specifically, unpaired two-sided Student’s *t*-tests were used to determine the significance of the differences (*, *p* < 0.05; **, *p* < 0.01; ***, *p* < 0.001).

## 5. Conclusions

In this work, the interaction between GmSNF1 and GmPKS4 was identified by the yeast two-hybrid system and BiFC technique. The *GmSNF1* gene responded positively to some abiotic stresses and its encoded protein localized in the nucleus and cytoplasm. Heterologous expression of the *GmSNF1* gene in a yeast system enhanced salt and salt–alkali tolerance of yeast strains. Then, the function of the *GmSNF1* gene in transgenic *Arabidopsis* and soybeans was analyzed. Overexpression of *GmSNF1* in plants can increase the expression of the CAT enzyme synthesis gene and some key stress-related genes, enhance CAT enzyme activity, and improve salt and salt–alkali tolerance. This research lays the foundation for further study on the signaling pathways involving *GmSNF1* and indicates that *GmSNF1* can be used as a valuable gene resource for plant breeding.

## Figures and Tables

**Figure 1 ijms-24-12482-f001:**
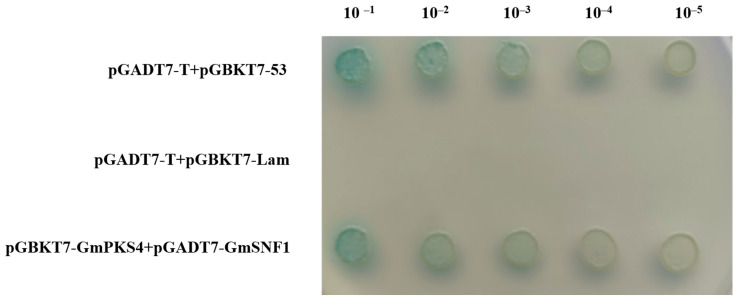
Validation of GmPKS4 and GmSNF1 proteins with yeast two-hybrid assay. The pGBKT7-GmPKS4 and pGADT7-GmSNF1 plasmids were transformed into Y2HGold yeast cells. The transformants were then inoculated on SD medium lacking Leu, Trp, and histidine (His) but containing X-α-Gal (SD/-Leu/-Trp/-His/X-α-Gal) and incubated at 30 °C for 5 days. Blue colonies indicate interactions. Co-transformations with pGADT7-T+ pGBKT7-53, pGADT7-T+ pGBKT7-Lam were used as a positive control and a negative control, respectively.

**Figure 2 ijms-24-12482-f002:**
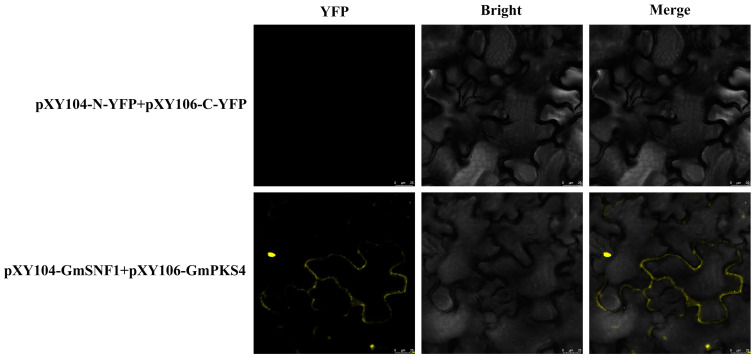
Interaction between GmSNF1 and GmPKS4 verified by BiFC system. Bimolecular fluorescence complementation assay: GmSNF1-YFP and GmPKS4-YFP were transiently expressed in tobacco leaves for 48 h. The YFP signal was observed under confocal microscopy. Scale bar = 25 μm. BiFC = bimolecular fluorescence complementation.

**Figure 3 ijms-24-12482-f003:**
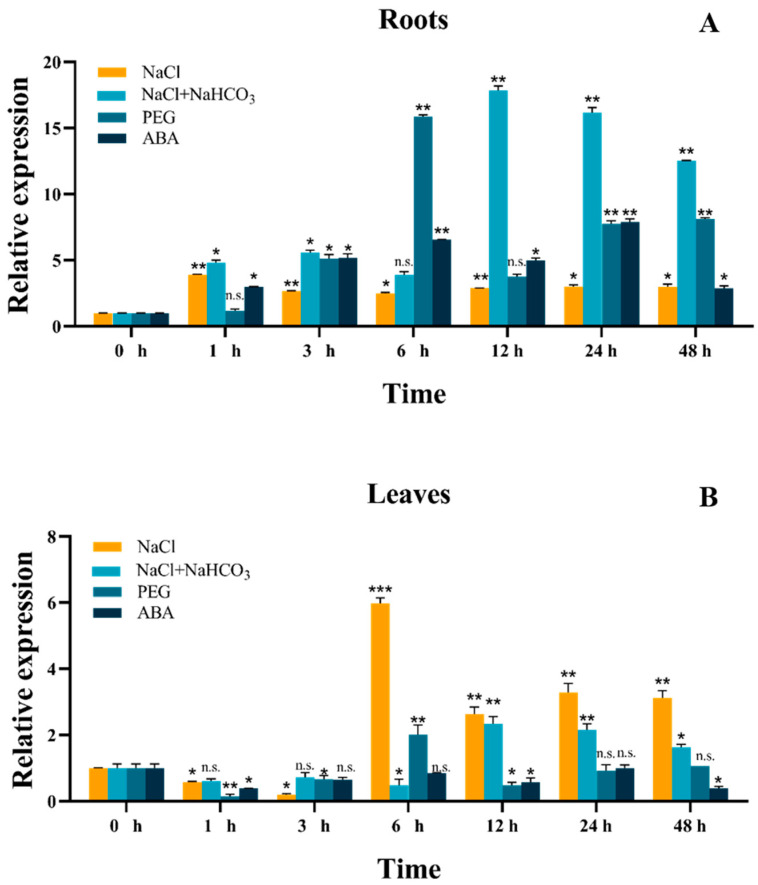
Expression of *GmSNF1* gene in soybean under abiotic stresses. (**A**) Relative expression of *GmSNF1* gene in soybean roots under abiotic stress treatment. (**B**) Relative expression of *GmSNF1* gene in soybean leaves under abiotic stress treatment. Data represent mean and standard deviation of three repeats (n = 3). Significant differences were determined by unpaired two-sided Student’s *t*-tests (*, *p* < 0.05; **, *p* < 0.01; ***, *p* < 0.001; n.s., not statistically significant).

**Figure 4 ijms-24-12482-f004:**
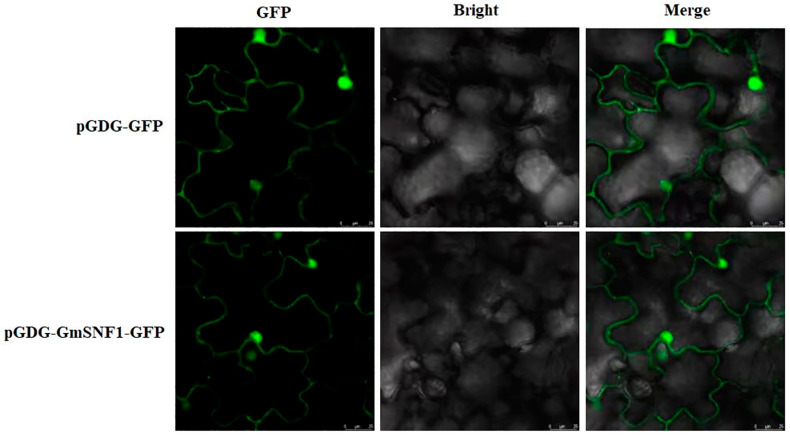
Subcellular localization analysis of soybean GmSNF1 protein. The GmPKS4-GFP fusion construct was obtained by inserting the complete GmSNF1 coding sequence (without TAG) at the *Xho* I and *Sal* I sites of pGDG-GFP. The pGDG-GmSNF1-GFP or pGDG-GFP vector (control) was infiltrated by *Agrobacterium rhizogenes* into young leaves of 4-week-old tobacco plants to observe protein localization in leaves 48 h after infiltration using confocal laser scanning microscopy G (Zeiss, Germany) [21]. Scale bar = 25 μm.

**Figure 5 ijms-24-12482-f005:**
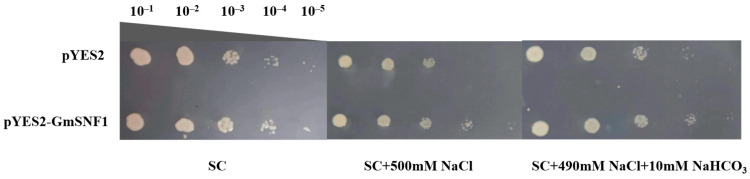
Effect of *GmSNF1* overexpression on the tolerance of yeast strains of INVSc1 to salt and salt–-alkali stress. pYES2 represents the yeast with an empty vector, and pYES2–GmSNF1 represents the yeast overexpressing *GmSNF1*. Photos were taken after 72 h of incubation at 30 °C. The dilution rates of SC medium containing the yeast strains were 10^−1^, 10^−2^, 10^−3^, 10^−4^, and 10^−5^.

**Figure 6 ijms-24-12482-f006:**
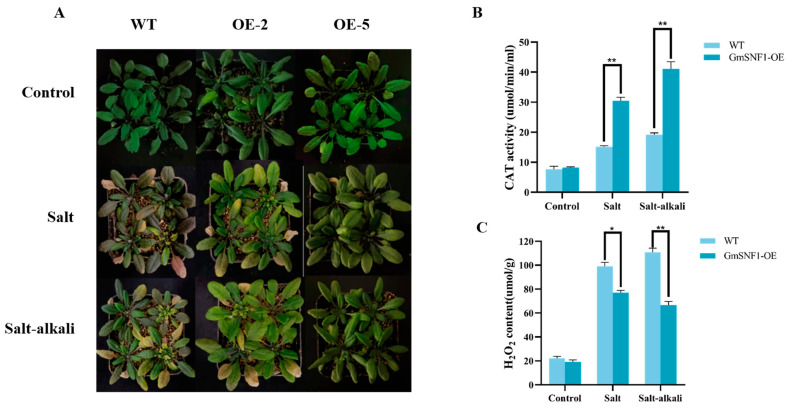
Phenotypic analysis and determination of CAT activity and hydrogen peroxide content in *Arabidopsis thaliana* under salt and salt-alkali stress. The wild-type (WT) *Arabidopsis* is Columbia 0. GmSNF1-OE, OE-2, and OE-5 are transgenic *Arabidopsis* overexpressing *GmSNF1.* (**A**) Phenotype of *Arabidopsis thaliana* under salt and salt-alkali stress. (**B**) CAT activity in *Arabidopsis thaliana* under salt and salt-alkali stress. (**C**) Accumulation of hydrogen peroxide in *Arabidopsis thaliana* under salt and salt-alkali stress. Significant differences were tested by unpaired two-sided Student’s *t*-tests (*, *p* < 0.05; **, *p* < 0.01).

**Figure 7 ijms-24-12482-f007:**
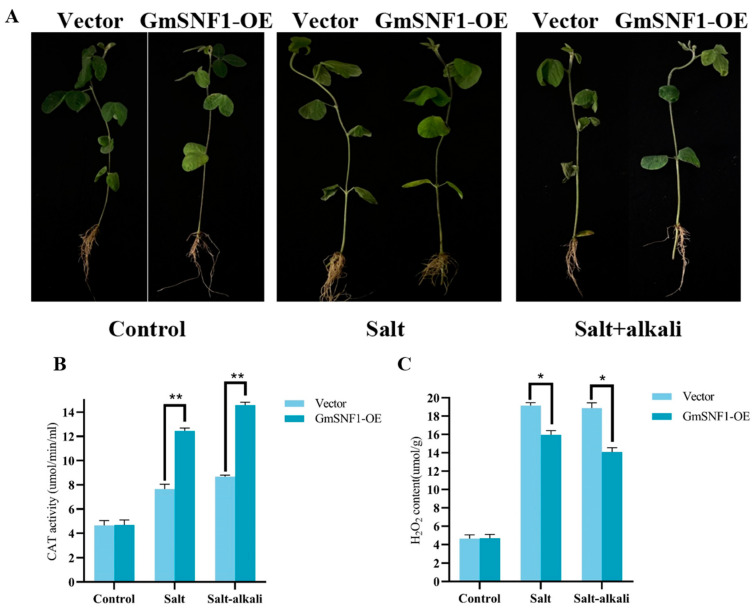
Phenotypic analysis of GmSNF1-OE soybeans under salt and salt-alkali stress. (**A**) Phenotype of GmSNF1-OE soybeans under salt and salt-alkali stress. (**B**) CAT activity of GmSNF1-OE soybeans under salt and salt-alkali stress. (**C**) Hydrogen peroxide content of GmSNF1-OE soybeans under salt and salt-alkali stress. Significant differences were tested by unpaired two-sided Student’s *t*-tests (*, *p* < 0.05; **, *p* < 0.01).

**Figure 8 ijms-24-12482-f008:**
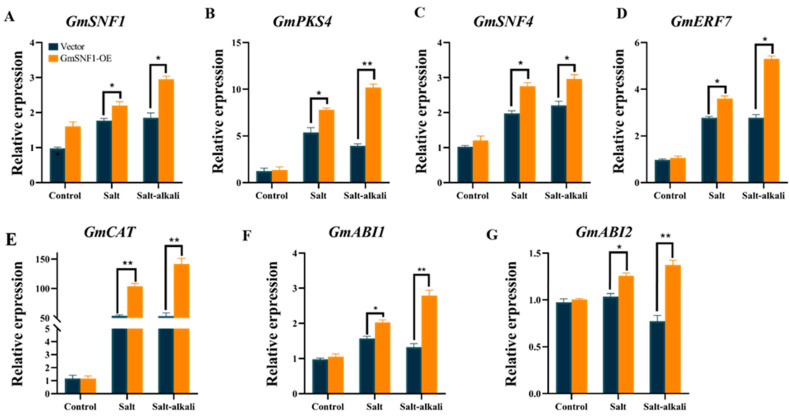
Relative transcript levels of some stress-related genes in GmSNF1-OE soybean plants under salt and salt-alkali stress. (**A**) Expression of *GmSNF1* in GmSNF1-OE soybean plants. (**B**) Expression of *GmPKS4* in GmSNF1-OE soybean plants. (**C**) Expression of *GmSNF4* in GmSNF1-OE soybean plants. (**D**) Expression of *GmERF7* in GmSNF1-OE soybean plants. (**E**) Expression of *GmCAT* in GmSNF1-OE soybean plants. (**F**) Expression of *GmABI1* in GmSNF1-OE soybean plants. (**G**) Expression of *GmABI2* in GmSNF1-OE soybean plant. Data represent mean and standard deviation of three repeats (n = 3). Significant differences were tested by unpaired two-sided Student’s *t*-tests (*, *p* < 0.05; **, *p* < 0.01).

**Figure 9 ijms-24-12482-f009:**
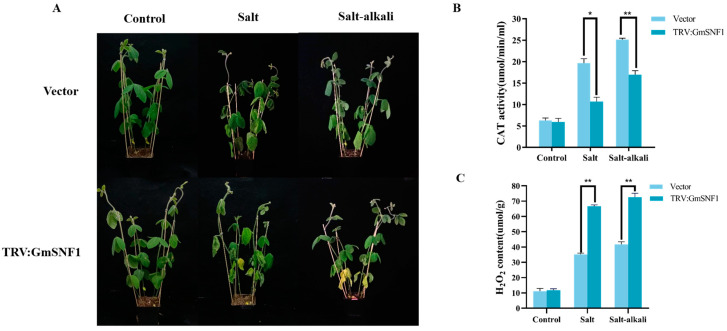
Responses of *GmSNF1* gene-silenced soybean plants to salt and salt-alkali stresses. (**A**) Phenotype of *GmSNF1* gene-silenced soybean under salt and salt-alkali stress. (**B**) CAT activity of *GmSNF1* gene-silenced soybean under salt and salt-alkali stress. (**C**) Hydrogen peroxide accumulation in *GmSNF1* gene-silenced soybean under salt and salt–alkali stress. Significant differences were tested by unpaired two-sided Student’s *t*-tests (*, *p* < 0.05; **, *p* < 0.01).

**Figure 10 ijms-24-12482-f010:**
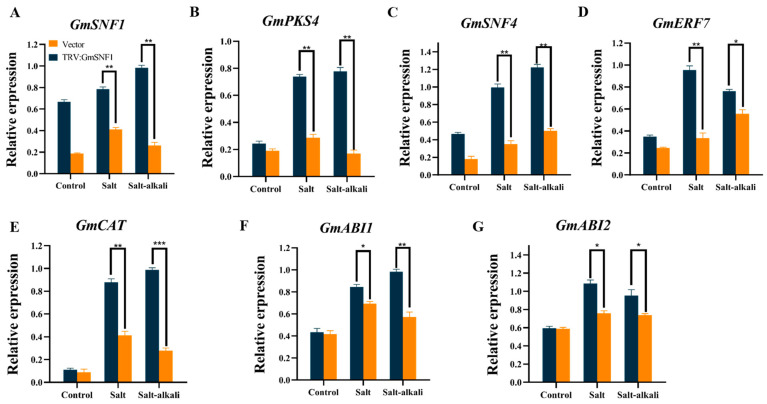
Relative transcript levels of some stress-related genes in *GmSNF1* gene-silenced soybean plants. (**A**) *GmSNF1* gene expression in *GmSNF1* gene-silenced soybean plants. (**B**) *GmPKS4* gene expression in *GmSNF1* gene-silenced soybean plants. (**C**) *GmSNF4* gene expression in *GmSNF1* gene-silenced soybean plants. (**D**) *GmERF7* gene expression in *GmSNF1* gene-silenced soybean plants. (**E**) *GmCAT* gene expression in *GmSNF1* gene-silenced soybean plants. (**F**) *GmABI1* gene expression in *GmSNF1* gene-silenced soybean plants. (**G**) *GmABI2* gene expression in *GmSNF1* gene-silenced soybean plants. Data represent mean and standard deviation of three repeats (n = 3). Significant differences were tested by unpaired two-sided Student’s *t*-tests (*, *p* < 0.05; **, *p* < 0.01; ***, *p* < 0.001).

## Data Availability

Data is unavailable due to privacy.

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
