# Peer review of "A Soybean Sucrose Non-Fermenting Protein Kinase 1 Gene, GmSNF1, Positively Regulates Plant Response to Salt and Salt–Alkali Stress in Transgenic Plants"

_ijms, 2023, doi:10.3390/ijms241512482_

Round 1
Reviewer 1 Report
No attached supplementary file by the Authors, although they mention it in the manuscript (Figure S1; Figure S2;Figure S3;TableS1). Lack of substantial figures and table should result in rejection of the manuscript.
Extensive editing of English language is required.
Author Response
Thank you very much for reviewing our manuscript. We have carefully corrected manuscript as per your instructions. Please find below a point-by-point response to each comment.
Major comments: No attached supplementary file by the Authors, although they mention it in the manuscript (Figure S1; Figure S2; Figure S3; TableS1). Lack of substantial figures and table should result in rejection of the manuscript.
Response: Thank you very much for highlighting this point. Due to our careless submission, we omitted to upload the supplementary data. We have added these files during revised submission.
Comment: Extensive editing of English language is required.
Response: Thank you very much for this suggestion. We would like to extensive English language editing with ZTEDIT. And all corrections in revised MS are highlighted in yellow.

Reviewer 2 Report
-There are a lot of mistakes in adjacent writing. Please separate citations and some adjacent sentences. You can use free AI programs like Grammarly for punctuation and syntax errors.
-Line 164: Arabidopsis should be italic.
-Line 170: Catalase should not start with an uppercase letter. Please write as catalase (CAT).
-Fig 6 . What is WT? There is no information about the WT in the caption of the figure. Figures and tables must be self-explanatory.
-LÄ°ne 331. You already abbreviated CAT before! Please be aware of the use of abbreviations throughout the manuscript.
The English quality is not bad. There are some syntax and punctuation errors. After minor revisions it is ok.
Author Response
Thank you very much for reviewing our manuscript. We have carefully corrected manuscript as per your instructions. Please find below a point-by-point response to each comment.
Point1: There are a lot of mistakes in adjacent writing. Please separate citations and some adjacent sentences. You can use free AI programs like Grammarly for punctuation and syntax errors.
Response: Thank you very much for this suggestion. We agree with you and have carefully corrected all these mistakes you mentioned. Please refer to changes with highlight.
Point2: Line 164: Arabidopsis should be italic.
Response: Thank you for this clarification. We checked the full text and corrected all similar errors. Please refer to the corrections with highlight.
Point3: Line 170: Catalase should not start with an uppercase letter. Please write as catalase (CAT).
Response: Thank you for this clarification. The change has been done. Please refer to line187.
Point4: Fig 6. What is WT? There is no information about the WT in the caption of the figure. Figures and tables must be self-explanatory.
Response: Thank you for pointing out this issue. WT means wild-type Arabidopsis. The change has been added to the caption of Figure 6.
Point5: Line 331. You already abbreviated CAT before! Please be aware of the use of abbreviations throughout the manuscript.
Response: According to your valuable suggestion, changes have been made. Please refer to line193, line377. And some other similar mistakes have been changed also. Please refer to BiFC in line95, ABA in line218 and line426.
Reviewer 3 Report
The authors provided a manuscript on the GmSNF1 candidate gene regulating the response to salt stress. Salt stress is one of the most common abiotic stresses, in response to which dozens of genes in plants respond. The authors conducted a complex study on yeast, tobacco, Arabidopsis thaliana, using a variety of methods. There is no doubt about the results obtained. Unfortunately, there is no Conclusions section in the manuscript.
They have small comments.
Fig. 2 is very dark. Which fluorosonde was used? At what wavelength were the pictures taken?
143- the type of tobacco is not specified (and in the Methods as well)
396- the reference is not given according to the requirements of the journal
, carefully check the design of references, a lot without pages, without a volume, issue according to the requirements of the journal
Author Response
Thank you very much for reviewing our manuscript. We have carefully corrected manuscript as per your instructions. Please find below a point-by-point response to each comment.
Poit1: The authors provided a manuscript on the GmSNF1 candidate gene regulating the response to salt stress. Salt stress is one of the most common abiotic stresses, in response to which dozens of genes in plants respond. The authors conducted a complex study on yeast, tobacco, Arabidopsis thaliana, using a variety of methods. There is no doubt about the results obtained. Unfortunately, there is no Conclusions section in the manuscript.
Response: Thank you for this suggestion. We had added conclusions in revised MS. Please refer to Conclusions in revised MS.
Poit2: Fig. 2 is very dark. Which fluorosonde was used? At what wavelength were the pictures taken?
Response: Thank you for pointing out this issue. We replaced Figure2 with a new one. YFP fluorescence was analyzed using confocal laser scanning microscopy with excitation and emission wavelengths of 488 and 507 nm, respectively. The change has been done. Please refer to Method 4.2.
Poit3: 143- the type of tobacco is not specified (and in the Methods as well)
Response: Thank you for pointing out this issue. The type of tobacco is Benthamite tobacco. All the missing points have been added. Please refer to line95, line144, line451.
Poit4: 396- the reference is not given according to the requirements of the journal, carefully check the design of references, a lot without pages, without a volume, issue according to the requirements of the journal
Response: Thank you for pointing out this issue. We have added this reference as [47]. The whole part of References has been checked and corrected. Please refer to References in revised version of manuscript.
In addition, some of other necessary changes have been corrected. Such as valid email addresses of authors, caption of some figures and mistakes of the use of abbreviations have been modification. Please refer to changes with highlight in revised version of manuscript.
Round 2
Reviewer 1 Report
The study was focused on the role of soybean sucrose non-fermenting protein kinase 1 gene, GmSNF1, in regulation of plant response to salt and salt-alkali stress in transgenic plants. The Authors stated that using the yeast two-hybrid technique and BiFC technique, the GmSNF1 protein was shown to interact with the GmPKS4 protein. The GmSNF1 gene responded positively to salt and salt-alkali stress using qRT-PCR analysis, and the GmSNF1 protein was localized in the nucleus and cytoplasm using subcellular localization assay. The GmSNF1 gene was then heterologous expressed in yeast, and the GmSNF1 gene was tentatively identified as having salt and salt-alkali tolerance function. The results indicated that GmSNF1 might be useful in genetic engineering to improve plant salt and salt-alkali tolerance.
The paper is quite interesting, however, I recommend some improvements:
- Figure 3: Factorial ANOVA with subsequent post-hoc test (e.g. Tukey’s test) should be re-calculated, because there were two indicators tested (root/leaves and time course from 0 to 48 h).
- Figures 6-10: there is no statistical test in the caption of the figure.
- I recommend including the electropherograms presenting the RNA bands in agarose gels in the manuscript or in the Supplementary file – it would provide information regarding quality of RNA samples,
- The Authors used SYBR Green fluorescent dye during RT-PCR gene expression studies, hence, it is obligatory to perform Melting Curve Analysis, and results of this examination should be added in the manuscript or Supplementary file (e.g., JPG or TIFF file),
- Minor editing of English language is required.
Minor editing of English language is required.
Author Response
Thank you very much for reviewing our manuscript and putting forward constructive feedback. We have carefully corrected manuscript as per your instructions. Please find below a point-by-point response to each comment.
Point1: Figure 3: Factorial ANOVA with subsequent post-hoc test (e.g. Tukey’s test) should be re-calculated, because there were two indicators tested (root/leaves and time course from 0 to 48 h).
Response: Thank you very much for highlighting this problem. We have re-calculated these data by unpaired two-sided Student's t-tests. Please refer to the caption of Figure 3 and Materials and Methods 4.12 with highlight in green.
Point2: Figures 6-10: there is no statistical test in the caption of the figure.
Response: Thank you for pointing out this issue. We have added the statistical test in the caption of these Figures 6-10 with highlight in green.
Point3: I recommend including the electropherograms presenting the RNA bands in agarose gels in the manuscript or in the Supplementary file – it would provide information regarding quality of RNA samples,
Response: Thank you for putting forward this issue. We have supplemented some of the electrophoretic results of RNA. Please refer to Figures S7-S10.
Point4: The Authors used SYBR Green fluorescent dye during RT-PCR gene expression studies, hence, it is obligatory to perform Melting Curve Analysis, and results of this examination should be added in the manuscript or Supplementary file (e.g., JPG or TIFF file),
Response: Thank you for your valuable suggestion. We have added the Melting Curve Analysis. Please refer to Figure S1, Figure S4, and Figure S6.
Point5: Minor editing of English language is required.
Response: Thank you for your valuable suggestion. We have done some modification with highlight in green in revised version 2 of manuscript.
Round 3
Reviewer 1 Report
The revision was properly concucted. The manuscript may be considered for publication.